REGISTERED REPORT

# Registered report: Systematic identification of genomic markers of drug sensitivity in cancer cells

**John P Vanden Heuvel[1,2], Jessica Bullenkamp[3], Reproducibility Project: Cancer Biology***

[1]Indigo Biosciences, State College, United States; [2]Veterinary and Biomedical Sciences, Penn State University, University Park, PA, United States; [3]King's College London, London, United Kingdom

*For correspondence: tim@cos.io

**Group author details:**
Reproducibility Project: Cancer Biology See page 18

**Abstract** The Reproducibility Project: Cancer Biology seeks to address growing concerns about the reproducibility in scientific research by conducting replications of selected experiments from a number of high-profile papers in the field of cancer biology. The papers, which were published between 2010 and 2012, were selected on the basis of citations and Altmetric scores (*Errington et al., 2014*). This Registered Report describes the proposed replication plan of key experiments from "Systematic identification of genomic markers of drug sensitivity in cancer cells" by Garnett and colleagues, published in *Nature* in 2012 (*Garnett et al., 2012*). The experiments to be replicated are those reported in Figures 4C, 4E, 4F, and Supplemental Figures 16 and 20. Garnett and colleagues performed a high throughput screen assessing the effect of 130 drugs on 639 cancer-derived cell lines in order to identify novel interactions for possible therapeutic approaches. They then tested this approach by exploring in more detail a novel interaction they identified in which Ewing's sarcoma cell lines showed an increased sensitivity to PARP inhibitors (Figure 4C). Mesenchymal progenitor cells (MPCs) transformed with the signature *EWS-FLI1* translocation, the hallmark of Ewing's sarcoma family tumors, exhibited increased sensitivity to the PARP inhibitor olaparib as compared to MPCs transformed with a different translocation (Figure 4E). Knockdown mediated by siRNA of *EWS-FLI1* abrogated this sensitivity to olaparib (Figure 4F). The Reproducibility Project: Cancer Biology is a collaboration between the Center for Open Science and Science Exchange, and the results of the replications will be published by *eLife*.

## Introduction

In their 2012 *Nature* paper, Garnett and colleagues implemented a large-scale high throughput *in vitro* screen designed to assess interactions between drugs and cancer-derived human cell lines (*Garnett et al., 2012*). This study leveraged a collection of over 600 cell lines screened across 130 drugs, with the aim to uncover new interactions between known cancers and known drugs in order to identify new potential therapeutic avenues using extant drugs. They captured a large number of known gene-drug interactions of clinically active drugs and identified several novel gene–drug associations. The ability to accurately capture a large number of known clinically relevant drug response biomarkers as well as preferential cancer type sensitivies known to occur in the clinic, such as decreased sensitivity to BRAF inhibitors in *BRAF* mutant colorectal cancers relative to melanomas, demonstrated the effectiveness of this large-scale pharmacogenomic approach. A similar approach of interrogating a large panel of human cancer cell lines of diverse lineages to predict drug sensitivity was conducted and reported by Barretina and colleagues at the same time (*Barretina et al., 2012*).

Garnett and colleagues identified an unexpected highly significant association between the *EWS-FLI1* translocation and sensitivity to the PARP inhibitor olaparib (*Garnett et al., 2012*). The *EWS-FLI1* translocation is a defining cytogenetic characteristic of Ewing's sarcoma family tumors (ESFTs). ESFTs are highly malignant tumors that occur in the bone and soft tissue, usually in children. The translocation event combines part of the EWS protein to a member of the *ETS* transcription factor family; in 90% of cases, this is FLI1. This creates a novel transcription factor, EWS-FLI1, whose oncogenic actions and mechanisms are still being fully explored. The translocation event is thought to be the initiating event for the development of ESFTs (*Erkizan et al., 2010*; *Lessnick and Ladanyi, 2012*).

PARP1 has diverse functions in chromatin modification, mitosis and cell death, but it is most well studied in the context of DNA repair and transcriptional regulation (*Sonnenblick et al., 2014*). PARP1 is a key component of single stranded break (SSB) repair; however, loss of PARP1 activity can be compensated for through DNA repair via homologous recombination (HR). This makes PARP1 an interesting therapeutic target in the context of malignancies with deficient HR, such as BRCA1 and BRCA2 mutant breast and ovarian cancers. In these cancers, loss of PARP activity results in synthetic lethality; with both SSB and HR impaired, the accumulation of DNA damage eventually kills the tumor cells (*Jiang et al., 2015*; *Lord et al., 2015*; *Sonnenblick et al., 2014*). PARP inhibitors (PARPi), such as olaparib, are now at the forefront of treatment for breast and ovarian cancers, as well as other malignancies (*Feng et al., 2015*).

In Figure 4C, a predicted interaction between Ewing's sarcoma cells and the PARP inhibitor olaparib was tested. PARP inhibitors target BRCA-deficient cells that rely on alternative DNA damage repair pathways involving PARP. A panel of cell lines representing Ewing's sarcoma, a BRCA-deficient line, as well as other osteosarcomas and cancers of soft tissue and epithelium were treated with a range of concentrations of olaparib. The concentration of olaparib required to reduce colony formation by 90% or more was much less for Ewing's sarcoma cells (on par with the concentration required for the BRCA-deficient cell line) than for the non-Ewing's sarcoma cell lines. This experiment will be replicated in Protocol 1.

In Figure 4E, the hypothesis that mouse mesenchymal progenitor cells (MPCs) that had been transformed with the *EWS-FLI1* translocation would confer sensitivity to olaparib was tested. The sensitivity of these cells to olaparib were compared to MPCs transformed with a related translocation (*FUS-CHOP*) as well as to SK-N-MC cells, which have the *EWS-FLI1* translocation endogenously. Treatment with olaparib did not inhibit the viability of the *FUS-CHOP* transformed MPCs, but did inhibit the viability of the SK-N-MC cells. Olaparib also inhibited the viability of the *EWS-FLI1* transformed MEFs compared to the *FUS-CHOP* translocation. This experiment will be replicated in Protocol 2.

In Figure 4F, the effects of *EWS-FLI1* depletion on a cell line carrying the translocation endogenously was tested. A673 cells were transfected with siRNAs targeting *EWS-FLI1*, which resulted in a partial rescue of sensitivity to olaparib compared to control siRNA transfected cells. This experiment will be replicated in Protocol 3.

A paper published at the same time as Garnett and colleagues' work also confirmed that Ewing's sarcoma cell lines were sensitive to treatment with PARP inhibitors (*Brenner et al., 2012*). In a previous paper, Brenner and colleagues reported that in prostate cancer PARP was a cofactor for wild-type *ETS* transcription factors, which makes up one half of the defining translocation-based fusion transcription factor of Ewing's sarcoma, and that PARPi treatment of *ETS* positive prostate cancers disrupted their growth (*Brenner et al., 2011*; *Legrand et al., 2013*). Based on this finding, they examined the role of PARP1 and PARPi in Ewing's sarcoma. Using immunoprecipitation, they detected a direct interaction between the EWS-FLI1 fusion transcription factor and PARP1 (*Brenner et al., 2012*). Further, they reported that transforming a cell line (in this case, PC3 cells) with the **EWS-FLI1** translocation conferred sensitivity to treatment with olaparib, and that siRNA mediated knockdown of **EWS-FLI1** inhibited transwell migration of ESFT derived cell lines, but not osteosarcoma cell lines (*Brenner et al., 2012*). Multiple groups have also reported the unique sensitivity of EWS-FLI1 carrying Ewing's sarcoma derived cell lines to olaparib (*Lee et al., 2013*; *Norris et al., 2014*; *Ordóñez et al., 2015*). Additional work then demonstrated that, similar to breast and ovarian cancers harboring **BRCA1/2** mutations, Ewing's sarcomas may also have defects in DNA repair mechanisms, rendering them sensitive to PARP inhibition (*Stewart et al., 2014*). This has led to the start of clinical trials treating Ewing's sarcoma patients with combination therapies targeting multiple DNA damage pathways and PARP inhibition. Results from a small scale nonrandomized phase II human trial failed to show clinical efficacy in patients with metastatic and/or recurrent

Ewing sarcoma treated with only olaparib (*Choy et al., 2014*), but other trials are underway to explore the efficacy of PARP inhibition in combination with chemotherapy.

## Materials and methods

Unless otherwise noted, all protocol information and references were derived from the original paper or information obtained directly from the authors.

### Protocol 1: Colony formation assay of Ewing's sarcoma cell lines with olaparib

This experiment assesses the sensitivity of Ewing's sarcoma cell lines to the PARP inhibitor olaparib. A colony formation assay will be performed with Ewing's sarcoma, osteosarcoma, and BRCA2-deficient and BRCA-proficient cells treated with a range of olaparib concentrations to determine the effective concentration (number of colonies reduced by at least 90%). This protocol replicates the experiment reported in Figure 4C and Supplemental Figure 16.

### Sampling

- The experiment will be performed with two replicates and each experiment will use 5 Ewing's sarcoma cell lines and 7 osteosarcoma cell lines for a power of 82%.
  - See Power calculations for details.
- The experiment will use the following cell lines:
  - Ewing's sarcoma cell lines:
    - A673
    - TC-71
    - SK-N-MC
    - CHLA-9
    - CHLA-10
  - Osteosarcoma cell lines:
    - U-2-OS
    - SJSA-1
    - SAOS-2
    - HOS
    - MG-63
    - 143B
    - G-292
  - BRCA2-deficient cell line: [positive control]
    - DoTc2-4510
  - BRCA-proficient cell line: [negative control]
    - MES-SA
- Each cell line will be treated with the following conditions:
  - Vehicle (DMSO)
  - 0.1 µM olaparib
  - 0.32 µM olaparib
  - 1 µM olaparib
  - 3.2 µM olaparib
  - 10 µM olaparib

### Materials and reagents

| Reagent | Type | Manufacturer | Catalog # | Comments |
|---|---|---|---|---|
| Olaparib | Inhibitor | Selleck Chemicals | S1060 | Source shared during communication with authors. |
| DMSO | Chemical | Sigma Aldrich | 472301 | Source shared during communication with authors. |
| Phosphate buffered saline (PBS) | Buffer | Gibco-Life Technologies | 10010-023 | Source shared during communication with authors. |
| Giemsa stain | Chemical | Sigma Aldrich | G5637 | Source shared during communication with authors. |
| Methanol | Chemical | Fisher Scientific | BP1105-4 | Source shared during communication with authors. |

*Continued on next page*

*Continued*

| Reagent | Type | Manufacturer | Catalog # | Comments |
|---|---|---|---|---|
| DoTc2-4510 cells | Cell line | ATCC | CRL-7920 | Original source not specified. |
| MES-SA cells | Cell line | ATCC | CRL-1976 | Original source not specified |
| U-2-OS cells | Cell line | ATCC | HTB-96 | Original source not specified. |
| SAOS-2 cells | Cell line | ATCC | HTB-85 | Original source not specified. |
| SJSA-1 cells | Cell line | ATCC | CRL-2098 | Original source not specified. |
| HOS cells | Cell line | ATCC | CRL-1543 | Original source not specified. |
| MG-63 cells | Cell line | ATCC | CRL-1427 | Original source not specified. |
| 143B cells | Cell line | ATCC | CRL-8303 | Replaces osteosarcoma cells used originally; see Known Differences. |
| G-292 cells, clone A141B1 | Cell line | ATCC | CRL-1423 | |
| A673 cells | Cell line | ATCC | CRL-1598 | Replaces the ES cells used originally; see Known Differences |
| SK-N-MC cells | Cell line | ATCC | HTB-10 | |
| TC-71 cells[2] | Cell line | Children's Oncology Group Cell Culture and Xenograft Repository | | |
| CHLA-10 cells[1] | Cell line | Children's Oncology Group Cell Culture and Xenograft Repository | | |
| CHLA-9 cells[3] | Cell line | Children's Oncology Group Cell Culture and Xenograft Repository | | |
| Iscove's modified DMEM (IMDM) | Cell culture | Life Technologies | 12440-053 | Not originally included. |
| L-glutamine | Cell culture | Life Technologies | 25030-081 | Not originally included. |
| Insulin-Transferrin-Selenium (ITS) | Growth factor | Lonza | 17-838Z | Not originally included. |
| McCoy's 5A Medium Modified | Cell culture | ATCC | 30-2007 | Not originally included. |
| Fetal bovine serum (FBS) | Cell culture | Valley Biomedical | BS3032 | Original source not specified. |
| RPMI 1640 medium | Cell culture | ATCC | 30-2001 | Original source not specified. |
| Eagle's Minimum Essential Media (EMEM) | Cell culture | ATCC | 30-2003 | Originally not specified. |
| 5-bromo-2'-deoxyuridine | Nucleoside | Sigma | B5002 | Not originally included. |
| MEM Eagle with Earle's BSS | Cell culture | Lonza | 12-125F | Not originally included. |
| DMEM – High Glucose | Cell culture | GE-Healthcare | E15-883 | Shared during communication with authors. |
| DMEM/F12 | Cell culture | Life Technologies | 11320-033 | Original source not specified. |

[1] See http://www.cogcell.org/dl/EFT_Lines_DataSheets/CHLA-10_Cell_Line_Data_Sheet_COGcell_org.pdf

[2] See http://www.cogcell.org/dl/EFT_Lines_DataSheets/TC- 71_Cell_Line_Data_Sheet_COGcell_org.pdf

[3] See http://www.cogcell.org/dl/EFT_Lines_DataSheets/CHLA- 9_Cell_Line_Data_Sheet_COGcell_org.pdf

## Procedure

Notes:

- All cell lines will be sent for STR profiling and mycoplasma testing.
- A673 cells are maintained in DMEM with 10% FBS.
- SAOS-2 are maintained in McCoy's 5A Medium Modified supplemented with 15% FBS.
- CHLA-10 cells and TC-71 are maintained in IMDM supplemented with 20% FBS, 4 mM L-glutamine, 5 µg/ml insulin, 5 µg/ml transferrin and 5 ng/ml selenium
- DoTc2-4510 cells are maintained in DMEM/F12 with 5% FBS.
- U-2-OS cells, HOS cells and G-292 cells are maintained in McCoy's 5A Medium Modified supplemented with 10% FBS.
- MG-63 cells are maintained in EMEM supplemented with 10% FBS.
- 143B cells are maintained in Minimum essential medium (Eagle) in Earle's BSS with 0.015 mg/ml 5-bromo-2'-deoxyuridine, 90%; FBS, 10%.
- SJSA-1 cells and SK-N-MC cells are maintained in RPMI 1640 medium supplemented with 10% FBS.

- MES-SA cells are maintained in McCoy's 5A Medium Modified supplemented with 10% FBS.
- All cells kept at 37°C and 5% $CO_2$.
- Olaparib is stored as a 10 mM stock in DMSO at -80°C. Each aliquot is subjected to no more than 5 freeze-thaw cycles.

1. Plate cells at low density in 6 well culture plates.
   a. Seed 2,000 cells per well in 2 ml of appropriate medium.
   b. Plate 6 wells per cell line in duplicate plates.
      i. Each cell line undergoes 6 treatments (see Sampling section above).
      ii. Label one plate A and one plate B for each cell line.
   c. Let cells adhere overnight.
2. The following day treat cells with varying concentrations of drug:
   a. Vehicle (DMSO at 0.1% v/v)
   b. 0.1 µM olaparib
   c. 0.32 µM olaparib
   d. 1 µM olaparib
   e. 2 µM olaparib
   f. 10 µM olaparib
3. Replace media and drug every 3-4 days.
4. After 7 to 21 days, when sufficient colonies are visible in the DMSO controls, fix cells for quantification.
   a. Stain cells once sufficient numbers of colonies are visible in DMSO wells.
      i. Sufficient colonies means at least 100 colonies, ideally over 200 colonies, are present in the vehicle treated wells for each cell line.
      ii. DoTc2-24510 cells were cultured for about 12 days in the original study.
   b. Wash cells once in PBS.
   c. Fix in ice-cold methanol for 30 min while gently shaking at room temperature.
   d. Remove methanol and add Giemsa stain at 1:20 dilution in deionized water. Incubate for 4 hr at room temperature shaking or overnight at 4° shaking.
   e. 4 hr later, or the following day, rinse cells with water and air dry.
5. Take brightfield images of plates and manually quantify the number of colonies, blinded, in each well from each plate.
6. Determine and record the concentration at which colony formation was reduced by >90% compared to DMSO controls for each plate.

## Deliverables

- Data to be collected:
  - Images of all plates
  - Colony counts of each well
  - Graph of each cell line and the concentration of olaparib required to reduce colony formation by >90% compared to DMSO controls. (Compare to Figure 4C)

## Confirmatory analysis plan

- Statistical Analysis of the Replication Data:
  - Wilcoxon-Mann-Whitney test for ordinal data of the effective concentration of olaparib to reduce the colonies by at least 90% in Ewing's sarcoma compared to osteosarcoma cell lines. Perform for each group (A or B) of replicate plates.
- Meta-analysis of original and replication attempt effect sizes:
  - This replication attempt will perform the statistical analysis listed above, compute the effect sizes (for each independent attempt), compare them against the reported effect size in the original paper and use a meta-analytic approach to combine the original and replication effects, which will be presented as a forest plot.

## Known differences from the original study

- The replication attempt will only examine Ewing's sarcoma and osteosarcoma derived cell lines, with the BRCA2-deficient cell line as a positive control, and will not include the remaining cell types (soft tissue and epithelial).

- Due to the inability to obtain any of the Ewing's sarcoma cell lines used originally, and in consultation with the original authors, the replication attempt will use A673, TC-71, CHLA-9, SK-N-MC and CHLA-10 cells. The cell lines all carry the critical EWS/FLI1 translocation. The cells used in the original study were ES1, ES6, ES7, ES8, and MHH-ES-1.
- Similarly, the replication attempt will use U-2-OS, SJSA-1, SAOS-2, HOS, MG-63, 143B, and G-292 cells. 143B and G-292 cells were not used in the original study and CAL-72, HuO-3N1, and NY cells that were used in the original study will not be included in this replication attempt.
- All known differences are listed in the materials and reagents section above with the originally used item listed in the comments section. All differences have the same capabilities as the original and are not expected to alter the experimental design.

## Provisions for quality control

The cell lines used in this experiment will undergo STR profiling to confirm identity and will be sent for mycoplasma testing to ensure there is no contamination. The DMSO concentration, although not originally reported, will be kept at a low percentage to avoid toxicity. All data obtained from the experiment will be made publicly available, either in the published manuscript or as an open access dataset available on the OSF (https://osf.io/nbryi/).

## Protocol 2: Olaparib sensitivity in cells transformed with the *EWS-FLI1* rearrangement

This experiment assesses if sensitivity to PARP inhibitors is due to the presence of the *EWS-FLI1* rearrangement. Mouse mesenchymal progenitor cells (MPCs) transformed with *EWS-FLI1*, or the related liposarcoma-associated translocation *FUS-CHOP*, will be analyzed for cellular viability after olaparib treatment. This protocol replicates the experiment reported in Figure 4E.

### Sampling

- The experiment will be repeated three times for a power of 99%.
  - See Power calculations for details.
- The experiment will use three cell lines:
  - *EWS-FLI1* transformed MPCs
  - *FUS-CHOP* transformed MPCs
  - SK-N-MC cells
    - These cells harbor the endogenous *EWS-FUS1* translocation
- Each cell line will be treated with the following conditions in technical triplicate:
  - No treatment [additional]
  - Vehicle (DMSO)
  - 0.39 µM olaparib
  - 0.78 µM olaparib
  - 1.56 µM olaparib
  - 3.13 µM olaparib
  - 6.25 µM olaparib
  - 12.5 µM olaparib

### Materials and reagents

| Reagent | Type | Manufacturer | Catalog # | Comments |
|---------|------|--------------|-----------|----------|
| EWS-FLI1 transformed mouse mesenchymal progenitor cells (MPCs) | Cell line | Authors | N/A | Provided by the Stamenkovic lab |
| FUS-CHOP transformed mouse mesenchymal progenitor cells (MPCs) | Cell line | Authors | N/A | Provided by the Stamenkovic lab |
| SK-N-MC cells | Cell line | ATCC | HTB-10 | Source shared during communication with authors. |
| Olaparib | Inhibitor | Selleck Chemicals | S1060 | Source shared during communication with authors. |

*Continued on next page*

*Continued*

| Reagent | Type | Manufacturer | Catalog # | Comments |
|---|---|---|---|---|
| DMSO | Chemical | Sigma | D8418 | Source shared during communication with authors. |
| 4% formaldehyde | Chemical | USB | 19943 | Source shared during communication with authors. |
| Syto60 fluorescent nucleic acid stain | Chemical | Invitrogen | S11342 | Catalog # shared during communication with authors. |
| FBS | Cell culture | Valley Biomedical | BS3032 | Original source not specified. |
| RPMI 1640 medium | Cell culture | ATCC | 30-2001 | Original source not specified. |
| Fluorescent plate reader | Equipment | LiCor | | Source shared during communication with authors. |
| DMEM, low glucose, GlutaMAX supplement, pyruvate | Cell culture | Gibco | 21885-025 | Shared during communication with authors. |
| MCDB 201 medium, with trace elements, L-glutamine and 30 mM HEPES; powder | Cell culture | Sigma | M6770 | Shared during communication with authors. |
| Ascorbic acid-2-phosphate | Cell culture | Sigma | A8960 | Shared during communication with authors. |
| Dexamethasone | Chemical | Sigma | D8893 | Shared during communication with authors. |
| Linoleic acid-BSA | Chemical | Sigma | L9530 | Shared during communication with authors. |
| Insulin, transferrin, sodium selenite supplement | Growth factor | Roche (Sigma) | 1074547 | Shared during communication with authors. |
| Dialyzed FCS | Cell culture | Sigma | F0392 | Shared during communication with authors. |
| EGF; human | Growth factor | Sigma | E9644 | Shared during communication with authors. |
| PDGF-BB, rat | Growth factor | R&D Systems | 520-BB-050 | Shared during communication with authors. |
| Penicillin-Streptomycin; 100X | Cell culture | Sigma | P4333 | Original source not specified. |
| Leukemia inhibitory factor (LIF); human; 10 µg/ml | Growth factor | Sigma | L5283 | Shared during communication with authors. Replaces LIF generated from CHO LIF720D cells. |
| Fibronectin; 0.1% in PBS | Chemical | Sigma | F1141 | Shared during communication with authors. |

## Procedure

- All cell lines will be sent for STR profiling and mycoplasma testing.
- SK-N-MC cells are maintained in RPMI-1640 with 10% FBS.
- MPCs are maintained in DMEM:MCDB (60:40) supplemented with 100 µM ascorbic acid-2-phosphate, 1 nM dexamethasone, 0.2 mg/ml linoleic acid-BSA, 5 µg/ml insulin, 5 µg/ml transferrin, 5 ng/ml sodium selenite, 2% dialyzed FCS, 10 ng/ml human EGF, 10 ng/ml rat PDGF-BB, 1X penicillin/streptomycin, and 10 ng/ml LIF. Coat culture dishes for cells with fibronectin (0.0001% in PBS) for 3 hr at 37°C (or 4°C overnight) before plating. Additional details available at: https://osf.io/2vxnj/?view_only=7c9fb185e4c64ae78660cad92083aaa1
- All cells are kept at 37°C and 5% $CO_2$.
- Olaparib is stored as a 10 µM stock in DMSO at -80°C. Each aliquot is subjected to no more than 5 freeze-thaw cycles.

1. Determine seeding density of each cell line so cells will be in the growth phase at the end of the assay (~70% confluency):
   a. Plate 500 – 1.6x10$^4$ *EWS-FLI1* transformed MPCs, *FUS-CHOP* transformed MPCs, and SK-N-MC cells in 96 well plates with 100 µl of appropriate medium in technical triplicate. Seed three plates for measurements at 48, 72, and 96 hr after seeding. Incubate overnight.

 b. 48 hr after seeding fix cells in 4% paraformaldehyde (PFA) for 30 min at 37°C.
 i. Stain cells with 1 µM Syto60 fluorescent nuclear dye, diluted in PBS, for 1 hr following manufacturer's instructions.
 1. Wash out excess Syto60 prior to signal reading.
 ii. Measure fluorescent signal intensity with a fluorescent plate reader.
 c. 24 hr later (72 hr after seeding) fix and stain cells with Cyto60 as described above and measure fluorescent signal intensity.
 d. 24 hr later (96 hr after seeding) fix and stain cells with Cyto60 as described above and measure fluorescent signal intensity.
 i. Use seeding density for each cell line that results in sub-confluency (~70%) at the end of the assay and where the signal is still in the linear range.
2. Seed cells at density determined in step 1 above in 96-well plates and let grow overnight.
 a. Seed 21 wells per cell line.
 i. Each cell line will be treated with 7 concentrations of drug in technical triplicate (see Sampling section above).
 b. Seed additional wells in technical triplicate per cell line for measurements at 24, 48, and 72 hr after treatment to test for proliferation of cells (no-treatment condition).
3. The next day, treat cells with a range of concentrations of olaparib.
 a. No-treatment [additional]
 b. Vehicle (DMSO at 0.1% v/v)
 c. 0.39 µM olaparib
 d. 0.78 µM olaparib
 e. 1.56 µM olaparib
 f. 3.13 µM olaparib
 g. 6.25 µM olaparib
 h. 12.5 µM olaparib
4. Incubate for 24, 48, or 72 hr.
 a. Medium does not need to be changed during this period.
 b. No-treatment wells are incubated for 24, 48, or 72 hr.
 c. Olaparib or vehicle treated wells are incubated for 72 hr.
5. After 24, 48, or 72 hr fix cells in 4% PFA for 30 min at 37°C.
6. Stain cells with 1 µM Syto60 fluorescent nuclear dye, diluted in PBS, for 1 hr following manufacturer's instructions.
 a. Wash out excess Syto60 prior to signal reading.
7. Measure fluorescent signal intensity with a fluorescent plate reader.
 a. Excitation wavelength: 630 nm
 b. Emission wavelength: 694 nm
8. Repeat steps 2–7 independently two additional times.

## Deliverables

- Data to be collected:
  - Raw data of fluorescent readout for all wells
  - Graph of normalized readings for each drug concentration compared to vehicle only control (Compare to Figure 4E)

## Confirmatory analysis plan

- Statistical Analysis of the Replication Data:
  Note: At the time of analysis, we will perform the Shapiro-Wilk test and generate a quantile-quantile plot to assess the normality of the data. We will also perform Levene's test to assess homoscedasticity. If the data appears skewed we will perform the appropriate transformation in order to proceed with the proposed statistical analysis. If this is not possible we will perform the planned comparisons using the equivalent non-parametric test.
  - One way ANOVA on $IC_{50}$ values of olaparib, determined by spline interpolation, of each cell line with the following planned comparisons using Fisher's LSD test:
    - EWS-FLI1 transformed MPCs vs. FUS-CHOP transformed MPCs
    - FUS-CHOP transformed MPCs vs. SK-N-MC cells
- Meta-analysis of original and replication attempt effect sizes:
  - This replication attempt will perform the statistical analysis listed above, compute the effect sizes, compare them against the reported effect size in the original paper and use a

meta-analytic approach to combine the original and replication effects, which will be presented as a forest plot.

## Known differences from the original study

- Commercially available LIF will be used in place of LIF generated from CHO LIF720D cells, as suggested by the original authors.
- All known differences are listed in the materials and reagents section above with the originally used item listed in the comments section. All differences have the same capabilities as the original and are not expected to alter the experimental design.

## Provisions for quality control

The cell lines used in this experiment will undergo STR profiling to confirm identity and will be sent for mycoplasma testing to ensure there is no contamination. The DMSO concentration, although not originally reported, will be kept at a low percentage to avoid toxicity. The seeding density of each cell line will be empirically determined prior to conducting the replicates so cells will be still be in the growth phase at the end of the assay. Measurements will be taken at 24, 48, and 72 hr after seeding from cells not treated with drug to test for proliferation of cells during the assay. All data obtained from the experiment will be made publicly available, either in the published manuscript or as an open access dataset available on the OSF (https://osf.io/nbryi/).

## Protocol 3: Olaparib sensitivity after depletion of EWS-FLI1 from A673 cells

This experiment assesses the sensitivity of PARP inhibitors to the presence of the *EWS-FLI1* rearrangement. *EWS-FLI1* specific siRNA will be used to deplete the fusion mRNA from A673 cells, which harbor the translocation endogenously, and cell viability after olaparib treatment will be assessed. This protocol replicates the experiment reported in Figure 4F and Supplemental Figure 20.

## Sampling

- The experiment will be repeated three times for a minimum power of 80%.
  - See Power calculations for details.
- The experiment has 2 cohorts:
  - Cohort1: siControl transfected A673 cells
  - Cohort 2: siEF1 transfected A673 cells
- Each cohort will be treated with the following conditions to assess cell viability in technical triplicate:
  - Untreated
  - 100 uM olaparib or DMSO equivalent
  - 33.33 uM olaparib or DMSO equivalent
  - 11.11 uM olaparib or DMSO equivalent
  - 3.704 uM olaparib or DMSO equivalent
  - 1.235 uM olaparib or DMSO equivalent
  - 0.412 uM olaparib or DMSO equivalent
  - 0.137 uM olaparib or DMSO equivalent
  - 0.046 uM olaparib or DMSO equivalent
  - 0.015 uM olaparib or DMSO equivalent
- Each cohort will be treated with the following conditions for qRT-PCR analysis:
  - 1.3 µM olaparib or DMSO equivalent
- Quantitative RT-PCR performed in technical triplicate for the following genes:
  - EWS-FLI1
  - RPLP0 (internal control)

## Materials and reagents

| Reagent | Type | Manufacturer | Catalog # | Comments |
|---|---|---|---|---|
| A673 cells | Cell line | ATCC | CRL-1598 | Source shared during communication with authors. |

*Continued on next page*

*Continued*

| Reagent | Type | Manufacturer | Catalog # | Comments |
|---|---|---|---|---|
| Olaparib | Inhibitor | Selleck Chemicals | S1060 | Source shared during communication with authors. |
| DMSO | Chemical | Sigma | D2650 | Source shared during communication with authors. |
| siEF1 | Nucleic acid | Qiagen | Custom order | 5'-GGCAGCAGAACCCUUCUUACG-3' |
| siCT control siRNA | Nucleic acid | Qiagen | SI03650318 | Catalog number shared during communication with authors. |
| Cell Titer 96 Aqueous One Solution Cell Proliferation Assay | Reporter assay | Promega | G3582 | |
| DMEM - High Glucose | Cell culture | GE-Healthcare | E15-883 | Shared during communication with authors. |
| FBS | Cell culture | Valley Biomedical | BS3032 | Original source not specified. |
| O-MEM | Cell culture | Gibco | 31985-062 | Shared during communication with authors. |
| 96 well tissue culture test plates | Labware | TPP | 92096 | Source shared during communication with authors. |
| Lipofectamine RNAiMAX | Cell culture | Life Technologies | 13778-150 | Shared during communication with authors. |
| High-capacity cDNA reverse transcription kit | Kit | Applied Biosystems | 4368814 | Shared during communication with authors. |
| NucleoSpin RNA II kit | Kit | Machery-Nagel | 740955.50 | Shared during communication with authors. |
| Power SYBR Green PCR mastermix | Kit | Applied Biosystems | 4367659 | Shared during communication with authors. |
| qPCR machine | Equipment | ABI/PRISM | 7500 | Shared during communication with authors. |
| EWS-FLI1 primers | Nucleic acid | Synthesis left to the discretion of the replicating lab and recorded later | | Sequence shared during communication with authors. |
| RPLP0 primers | Nucleic acid | | | Sequence shared during communication with authors. |
| GloMax Multi+ Detection System (spectrophotometer) | Equipment | Promega | 9311-011 | Shared during communication with authors. Replaces BMG FLUOstar OPTIMA microplate reader. |

## Procedure

Notes:

- All cell lines will be sent for STR profiling and mycoplasma testing.
- A673 cells are maintained in DMEM with 10% FBS.
- All cells are kept at 37°C and 5% $CO_2$.
- Olaparib is stored as a 10 mM stock in DMSO at -80°C. Each aliquot is subjected to no more than 5 freeze-thaw cycles.
- siRNA stocks kept at 20 µM; final siRNA concentration is 25 nM.

1. Seed cells for assays:
   a. For cell viability assay, plate 5000 A673 cells per well in 64 µl medium without antibiotics in a 96-well plate.
      i. Seed enough cells for each condition to be performed in technical triplicate.
   b. For qRT-PCR, plate $3 \times 10^4$ A673 cells per well of a 24 well plate in medium without antibiotics.
      i. This is a similar seeding density as the 96 well plate.
2. Immediately transfect cells with 25 nM siControl or siEF1 siRNAs using Lipofectamine RNAi-MAX with the cells in suspension. The following directions prepare enough transfection mixture for one 96-well plate. The amounts will be scaled accordingly to account for the plates used for the qRT-PCR analysis.

 a. Mix 12.17 µl of 20 µM siRNA stock with 962.1 µl of OptiMEM.

 b. Mix 18.26 µl of Lipfectamine RNAiMAX with 956 µl OptiMEM.

 c. Gently mix the two solutions together and incubate for 12 min at room temperature.

 d. Add 16 µl of transfection mixture per well to appropriate wells.

3. Immediately after siRNA transfection, treat cells with varying concentrations of olaparib or vehicle (DMSO).

 a. See Sampling section above for details; include untreated cells and cells treated with vehicle only

 b. Prepare a 500 µM stock of Olaparib by adding 30 µl of 10 mM stock to 570 µl of DMEM.

 c. Prepare a stock of DMSO by adding 30 µl of DMSO to 570 µl of DMEM.

 i. These will be used for the vehicle treated cells.

 d. For cell viability assay, dilute olaparib and DMSO in DMEM by three-fold serial dilution as outlined:

**Experimental wells**

| Control | Olaparib (µM) | | | | | | | | | Background |
|---|---|---|---|---|---|---|---|---|---|---|
| No drug | 100 | 33.33 | 11.11 | 3.704 | 1.235 | 0.412 | 0.137 | 0.046 | 0.015 | No cells |
| | DMSO (µL used in olaparib dilution) | | | | | | | | | |
| | 1 | 0.333 | 0.111 | 0.037 | 0.012 | 0.004 | 0.001 | $5\times10^{-4}$ | $2\times10^{-4}$ | |

**Vehicle only wells**

| | DMSO (µL/well, no olaparib) | | | | | | | | |
|---|---|---|---|---|---|---|---|---|---|
| | 1 | 0.333 | 0.111 | 0.037 | 0.012 | 0.004 | 0.001 | $5\times10^{-4}$ | $2\times10^{-4}$ |

 i. Add 20 µl of each dilution to appropriate wells. Final volume per well is 100 µl.

 e. For qRT-PCR, treat cells with 1.3 µM olaparib or equivalent volume of DMSO.

 i. Dilute 500 µM stock of olaparib or stock of DMSO to create 6.5 µM (5X working solution) in DMEM. Add to plate to achieve 1.3 µM olaparib or equivalent volume of DMSO (0.013%).

4. Incubate cells for 72 hr.

 a. Medium does not need to be changed during this time period.

5. Measure cell viability by using the Cell Titer 96 well aqueous one assay according to the manufacturer's instructions.

 a. Add 20 µl Cell Titer 96 Aqueous solution reagent per well containing 100 µl medium.

 b. Incubate plate at 37°C in humidified 5% CO2 for 4 hr.

 c. Record absorbance at 490 nm using a BMG FLUOstar OPTIMA microplate reader.

 i. Subtract average background (no cell) wells from each treated (olaparib or DMSO) well.

 ii. Normalize values to corresponding untreated (no drug or vehicle) wells for each cohort.

 iii. Determine $IC_{50}$ value for each cohort using normalized olaparib values.

6. qRT-PCR to confirm knockdown of EWS-FLI1 expression:

 a. Extract RNA with the NuceloSpin RNA II kit according to manufacturer's instructions.

 i. Record $A_{260}/A_{280}$ and $A_{260}/A_{230}$ ratios.

 b. Synthesize cDNA using 1 µg of RNA and the High-capacity cDNA reverse transcription kit according to the manufacturer's instructions.

 c. Perform qPCR using POWER SYBR Green PCR mastermix according to the manufacturer's instructions in technical triplicate.

 i. Primers:

 1. *EWS-FLI1*(forward):

 a. 5'-GCCAAGCTCCAAGTCAATATAGC-3'

 2. *EWS-FLI1*(reverse):

 a. 5'-GAGGCCAGAATTCATGTTATTGC-3'

 3. *RPLP0*(forward): Internal Control

 a. 5'-GAAACTCTGCATTCTCGCTTC-3'

 4. *RPLP0*(reverse): Internal Control

 a. 5'-GGTGTAATCCGTCTCCACAG-3'

 ii. Reaction conditions run on an ABI PRISM 7500.

1. 95℃ for 10 min
2. 40 cycles of:
   a. 95℃ for 15 s
   b. 60℃ for 1 min
3. Dissociation curve
   iii. Analyze with 7500 SDS software or equivalent.
   d. Calculate relative *EWS-FLI1* expression for each sample using *RPLP0* as internal standard.
7. Repeat independently two additional times.

## Deliverables

- Data to be collected:
  - Raw absorbance values for all wells.
  - Graph of absorbance corrected values for all concentrations of olaparib or DMSO normalized to untreated controls (as seen in Figure 4F).
  - $IC_{50}$ values for each cohort using normalized olaparib values.
  - Raw and normalized qRT-PCR data (as seen in Supplemental Figure 20).

## Confirmatory analysis plan

- Statistical Analysis of the Replication Data:
  Note: At the time of analysis, we will perform the Shapiro-Wilk test and generate a quantile-quantile plot to assess the normality of the data. We will also perform Levene's test to assess homoscedasticity. If the data appears skewed we will perform the appropriate transformation in order to proceed with the proposed statistical analysis. If this is not possible we will perform the planned comparisons using the equivalent non-parametric test.
  - o Viability assay:
    - Unpaired two-tailed *t*-test of olaparib $IC_{50}$ values of siControl transfected cells compared to siEF1 transfected cells.
  - qRT-PCR:
    - Two-way ANOVA of siControl and siEF1 transfected cells treated with or without olaparib with the following planned comparisons using the Bonferroni correction:
      - siControl transfected cells treated with DMSO compared to siEF1 transfected cells treated with DMSO.
      - siControl transfected cells treated with olaparib compared to siEF1 transfected cells treated with olaparib.
- Meta-analysis of original and replication attempt effect sizes:
  - This replication attempt will perform the statistical analysis listed above, compute the effect sizes, compare them against the reported effect size in the original paper and use a meta-analytic approach to combine the original and replication effects, which will be presented as a forest plot.

## Known differences from the original study

- All known differences are listed in the materials and reagents section above with the originally used item listed in the comments section. All differences have the same capabilities as the original and are not expected to alter the experimental design.

## Provisions for quality control

The cell line used in this experiment will undergo STR profiling to confirm identity and will be sent for mycoplasma testing to ensure there is no contamination. The sample purity ($A_{260/280}$ ratio) of the isolated RNA from each sample will be reported. All data obtained from the experiment will be made publicly available, either in the published manuscript or as an open access dataset available on the OSF (https://osf.io/nbryi/).

## Power calculations

For additional details on power calculations, please see analysis scripts and associated files on the Open Science Framework:
https://osf.io/j9bnk/

## Protocol 1

Summary of original data estimated from graph reported in Figure 4C

| Cell type | Cell line | Effective concentration (µM) |
| --- | --- | --- |
| Ewing's sarcoma | ES1 | 1 |
| | ES6 | 1 |
| | ES7 | 0.32 |
| | ES8 | 1 |
| | MHH-ES-1 | 0.32 |
| Osteosarcoma | CAL-72 | 10 |
| | HOS | 1 |
| | HuO-3N1 | 3.2 |
| | MG-63 | 3.2 |
| | NY | 3.2 |
| | SAOS-2 | 3.2 |
| | SJSA-1 | 10 |
| | U-2-OS | 10 |
| BRCA2-deficient | DoTc2-4510 | 0.32 |

### Test family

- Wilcoxon-Mann-Whitney test (ordinal data): alpha error = 0.05

### Power calculations

- Power calculations were performed with R software, version 3.2.2 (*R Development Core Team, 2015*).

| Group 1 | Group 2 | Effect size (Cliff's delta) | A priori power | Group 1 sample size | Group 2 sample size |
| --- | --- | --- | --- | --- | --- |
| Ewing's sarcoma | Osteosarcoma | 0.92500 | 81.8% | 5 | 7 |

## Protocol 2

Summary of original data reported in Figure 4E (shared by authors)

| Cell line | Concentration of olaparib (µM) | Mean | SD | N |
| --- | --- | --- | --- | --- |
| *EWS-FLI1* transformed MPCs | 0 | 1 | 0.06 | 3 |
| | 0.39 | 0.59 | 0.05 | 3 |
| | 0.78 | 0.53 | 0.09 | 3 |
| | 1.56 | 0.44 | 0.05 | 3 |
| | 3.13 | 0.34 | 0.05 | 3 |
| | 6.25 | 0.24 | 0.04 | 3 |
| | 12.5 | 0.22 | 0.04 | 3 |

*Continued on next page*

*Continued*

| Cell line | Concentration of olaparib (μM) | Mean | SD | N |
|---|---|---|---|---|
| *FUS-CHOP* transformed MPCs | 0 | 1 | 0.09 | 3 |
| | 0.39 | 1.06 | 0.01 | 3 |
| | 0.78 | 1.03 | 0.06 | 3 |
| | 1.56 | 1.11 | 0.08 | 3 |
| | 3.13 | 0.98 | 0.09 | 3 |
| | 6.25 | 0.59 | 0.07 | 3 |
| | 12.5 | 0.45 | 0.04 | 3 |
| SK-N-MC | 0 | 1 | 0.04 | 3 |
| | 0.39 | 0.66 | 0.04 | 3 |
| | 0.78 | 0.66 | 0.09 | 3 |
| | 1.56 | 0.50 | 0.01 | 3 |
| | 3.13 | 0.40 | 0.04 | 3 |
| | 6.25 | 0.30 | 0.05 | 3 |
| | 12.5 | 0.25 | 0.03 | 3 |

$IC_{50}$ values of olaparib, determined by spline interpolation.
Calculations performed with R software, version 3.2.2 (*R Development Core Team, 2015*).

| Cell line | Mean | SD | N |
|---|---|---|---|
| *EWS-FLI1* transformed MPCs | 1.0502 | 0.5363 | 3 |
| *FUS-CHOP* transformed MPCs | 7.7963 | 1.3024 | 3 |
| SK-N-MC | 1.5449 | 0.0505 | 3 |

## Test family

- Two-tailed *t* test, Wilcoxon-Mann-Whitney test, Fisher's LSD: alpha error = 0.05

Power Calculations performed with G*Power software, version 3.1.7 (*Faul et al., 2007*).

| Group 1 | Group 2 | Effect size *d* | A priori power | Group 1 sample size | Group 2 sample size |
|---|---|---|---|---|---|
| *EWS-FLI1* transformed MPCs | *FUS-CHOP* transformed MPCs | 6.77343 | 83.8%[1] | 2[1] | 2[1] |
| SK-N-MC | *FUS-CHOP* transformed MPCs | 6.78283 | 83.8%[1] | 2[1] | 2[1] |

[1] 3 samples per group will be used as a minimum making the power 99.9%.

## Test family

- Due to the large difference in variance, these parametric tests are only used for comparison purposes. The sample size is based on the non-parametric tests listed above.
- ANOVA: Fixed effects, omnibus, one-way: alpha error = 0.05.

Power Calculations performed with G*Power software, version 3.1.7 (*Faul et al., 2007*).
ANOVA F test statistic and partial $\eta^2$ performed with R software, version 3.2.2 (*R Development Core Team, 2015*).

| Groups | F test statistic | Partial η2 | Effect size f | A priori power | Total sample size |
|---|---|---|---|---|---|
| *EWS-FLI1* transformed MPCs, *FUS-CHOP* transformed MPCs, and SK-N-MC | F(2,6) = 64.06 | 0.95526 | 4.62097 | 99.9% | 6[1] (3 groups) |

[1] 9 total samples (3 per group) will be used as a minimum.

## Test family

- Due to the large difference in variance, these parametric tests are only used for comparison purposes. The sample size is based on the non-parametric tests listed above.
- Two-tailed *t* test, difference between two independent means, Fisher's LSD: alpha error = 0.05

Power Calculations performed with G*Power software, version 3.1.7 (*Faul et al., 2007*).

| Group 1 | Group 2 | Effect size *d* | A priori power | Group 1 sample size | Group 2 sample size |
|---|---|---|---|---|---|
| *EWS-FLI1* transformed MPCs | *FUS-CHOP* transformed MPCs | 6.77343 | 89.9%[1] | 2[1] | 2[1] |
| SK-N-MC | *FUS-CHOP* transformed MPCs | 6.78283 | 89.9%[1] | 2[1] | 2[1] |

[1] 3 samples per group will be used as a minimum making the power 99.9%.

## Protocol 3
### Viability assay
Summary of original data reported in Figure 4F (shared by authors)

| siRNA | Concentration of olaparib (μM) or volume of DMSO (μl) | Mean | SD | N |
|---|---|---|---|---|
| siControl (DMSO treatment) | 0 μl | 97.3360 | 0.95391 | 3 |
| | $2 \times 10^{-4}$ μl | 102.203 | 3.70013 | 3 |
| | $5 \times 10^{-4}$ μl | 100.088 | 0.90226 | 3 |
| | 0.001 μl | 94.8628 | 1.30022 | 3 |
| | 0.004 μl | 100.095 | 3.84743 | 3 |
| | 0.012 μl | 107.634 | 1.05370 | 3 |
| | 0.037 μl | 110.378 | 4.41561 | 3 |
| | 0.111 μl | 111.467 | 0.68191 | 3 |
| | 0.333 μl | 104.501 | 1.98400 | 3 |
| | 1.000 μl | 107.905 | 1.61184 | 3 |
| siControl (olaparib treatment) | 0 μM | 102.664 | 2.82201 | 3 |
| | 0.0152 μM | 95.9921 | 1.18048 | 3 |
| | 0.046 μM | 83.1889 | 2.80989 | 3 |
| | 0.1371 μM | 81.8370 | 2.93976 | 3 |
| | 0.411 μM | 72.4056 | 3.10030 | 3 |
| | 1.234 μM | 54.9026 | 2.74523 | 3 |
| | 3.70 μM | 16.0636 | 3.50915 | 3 |
| | 11.11 μM | 1.28032 | 0.61000 | 3 |
| | 33.33 μM | -1.45527 | 1.64101 | 3 |
| | 100 μM | 2.28231 | 2.39427 | 3 |

*Continued on next page*

*Continued*

| siRNA | Concentration of olaparib (μM) or volume of DMSO (μl) | Mean | SD | N |
|---|---|---|---|---|
| siEF1 (DMSO treatment) | 0 μl | 99.0971 | 1.13436 | 3 |
| | $2\times10^{-4}$ μl | 99.6397 | 1.21598 | 3 |
| | $5\times10^{-4}$ μl | 95.3622 | 0.45115 | 3 |
| | 0.001 μl | 90.4599 | 4.31934 | 3 |
| | 0.004 μl | 94.3179 | 0.86896 | 3 |
| | 0.012 μl | 95.1752 | 2.35064 | 3 |
| | 0.037 μl | 96.3837 | 1.39419 | 3 |
| | 0.111 μl | 96.7576 | 1.13467 | 3 |
| | 0.333 μl | 95.4762 | 1.38497 | 3 |
| | 1.000 μl | 97.2365 | 1.24839 | 3 |
| siEF1 (olaparib treatment) | 0 μM | 100.903 | 3.87004 | 3 |
| | 0.0152 μM | 97.9023 | 4.77067 | 3 |
| | 0.046 μM | 95.4853 | 5.47687 | 3 |
| | 0.1371 μM | 93.9713 | 2.33965 | 3 |
| | 0.411 μM | 89.4430 | 4.97093 | 3 |
| | 1.234 μM | 76.6332 | 2.436545 | 3 |
| | 3.70 μM | 45.0396 | 1.67473 | 3 |
| | 11.11 μM | 18.7815 | 1.78436 | 3 |
| | 33.33 μM | 11.7541 | 3.75220 | 3 |
| | 100 μM | 11.7997 | 2.22773 | 3 |

$IC_{50}$ values of olaparib, determined by four-parameter log-logistic function.
Calculations performed with R software, version 3.2.2 (*R Development Core Team, 2015*).

| Cell line | Mean | SD | N |
|---|---|---|---|
| siControl | 1.35191 | 0.0684 | 3 |
| siEF1 | 2.74561 | 0.1715 | 3 |

## Test family

- Two-tailed *t* test, Wilcoxon-Mann-Whitney test: alpha error = 0.05

Power Calculations performed with G*Power software, version 3.1.7 (*Faul et al., 2007*).

| Group 1 | Group 2 | Effect size *d* | A priori power | Group 1 sample size | Group 2 sample size |
|---|---|---|---|---|---|
| siControl | siEF1 | 10.67494 | 98.7%[1] | 2[1] | 2[1] |

[1] 3 samples per group will be used as a minimum making the power 99.9%.

## Test family

- Due to the large difference in variance, these parametric tests are only used for comparison purposes. The sample size is based on the non-parametric tests listed above.
- Two-tailed *t* test, difference between two independent means: alpha error = 0.05

Power Calculations performed with G*Power software, version 3.1.7 (*Faul et al., 2007*).

| Group 1 | Group 2 | Effect size $d$ | A priori power | Group 1 sample size | Group 2 sample size |
|---|---|---|---|---|---|
| siControl | siEF1 | 10.67494 | 99.6%[1] | 2[1] | 2[1] |

[1] 3 samples per group will be used as a minimum making the power 99.9%.

## qRT-PCR
Summary of original data estimated from graph reported in Supplemental Figure 20.

| Treatment | siRNA | Mean | SD | N |
|---|---|---|---|---|
| DMSO | siControl | 100 | 22.9 | 3 |
| | siEF1 | 4.75 | 0.838 | 3 |
| 1.3 µM olaparib | siControl | 90.5 | 14.5 | 3 |
| | siEF1 | 7.26 | 0.838 | 3 |

## Test family

- Two-tailed $t$ test, Wilcoxon-Mann-Whitney test, Bonferroni's correction: alpha error = 0.025

Power Calculations performed with G*Power software, version 3.1.7 (*Faul et al., 2007*).

| Group 1 | Group 2 | Effect size $d$ | A priori power | Group 1 sample size | Group 2 sample size |
|---|---|---|---|---|---|
| siControl cells treated with DMSO | siEF1 cells treated with DMSO | 5.87713 | 98.3% | 3 | 3 |
| siControl cells treated with olaparib | siEF1 cells treated with olaparib | 8.09108 | 99.9% | 3 | 3 |

## Test family

- Due to the large difference in variance, these parametric tests are only used for comparison purposes. The sample size is based on the non-parametric tests listed above.
- ANOVA: Fixed effects, special, main effects and interactions: alpha error = 0.05.

Power Calculations performed with G*Power software, version 3.1.7 (*Faul et al., 2007*).
ANOVA F test statistic and partial $\eta^2$ performed with R software, version 3.2.2 (*R Development Core Team, 2015*).

| Groups | F test statistic | Partial $\eta^2$ | Effect size $f$ | A priori power | Total sample size |
|---|---|---|---|---|---|
| A673 cells transfected with siControl or siEF1 and treated with DMSO or olaparib | F(1,8) = 129.85 (main effect: siRNA) | 0.94196 | 4.02877 | 99.2%[1] | 6[1] (4 groups) |

[1] 12 samples (3 per group) will be used based on the planned comparisons making the power 99.9%.

## Test family

- Due to the large difference in variance, these parametric tests are only used for comparison purposes. The sample size is based on the non-parametric tests listed above.
- 2 tailed $t$ test, difference between two independent means, Bonferroni's correction: alpha error = 0.025

Power Calculations performed with G*Power software, version 3.1.7 (*Faul et al., 2007*).

| Group 1 | Group 2 | Effect size d | A priori power | Group 1 sample size | Group 2 sample size |
|---|---|---|---|---|---|
| siControl cells treated with DMSO | siEF1 cells treated with DMSO | 5.87713 | 99.1% | 3 | 3 |
| siControl cells treated with olaparib | siEF1 cells treated with olaparib | 8.09108 | 80.6%[1] | 2[1] | 2[1] |

[1] 3 samples per group will be used based on the other comparisons making the power 99.9%.

## Acknowledgements

The Reproducibility Project: Cancer Biology core team would like to thank the original authors, in particular Dr. Cyril Benes and Dr. Ivan Stamenkovic, for generously sharing critical information as well as reagents to ensure the fidelity and quality of this replication attempt. We thank Courtney Soderberg at the Center for Open Science for assistance with statistical analyses. We would also like to thank the following companies for generously donating reagents to the Reproducibility Project: Cancer Biology; American Type and Tissue Collection (ATCC), Applied Biological Materials, BioLegend, Charles River Laboratories, Corning Incorporated, DDC Medical, EMD Millipore, Harlan Laboratories, LI-COR Biosciences, Mirus Bio, Novus Biologicals, Sigma-Aldrich, and System Biosciences (SBI).

## Additional information

### Group author details

Reproducibility Project: Cancer Biology

Elizabeth Iorns: Science Exchange, Palo Alto, United States; William Gunn: Mendeley, London, United Kingdom; Fraser Tan: Science Exchange, Palo Alto, United States; Joelle Lomax: Science Exchange, Palo Alto, United States; Nicole Perfito: Science Exchange, Palo Alto, United States; Timothy Errington: Center for Open Science, Charlottesville, United States

### Competing interests

JPVH: Indigo Biosciences is a Science Exchange associated laboratory. RP:CB: EI, FT, NP, and JL are employed and hold shares in Science Exchange, Inc. The other authors declare that no competing interests exist.

### Funding

| Funder | Author |
|---|---|
| Laura and John Arnold Foundation | Reproducibility Project: Cancer Biology |

The funder had no role in study design or the decision to submit the work for publication.

### Author contributions

JPVH, JB, Drafting or revising the article; RP:CB, Conception and design, Drafting or revising the article

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
