## [Decision Letter]

Thank you for submitting your work entitled "Registered report: Systematic identification of genomic markers of drug sensitivity in cancer cells" for consideration by *eLife*. Your article has been reviewed by three peer reviewers, and the evaluation has been overseen by a Reviewing Editor and Charles Sawyers as the Senior Editor. The reviewers have discussed the reviews with one another and the Reviewing Editor has drafted this decision to help you prepare a revised submission.

Summary:

This Registered report aims at assessing the reproducibility of the results of a large-scale pharmacogenomic study published in Nature in 2012. The study leverages a collection of over 600 cell lines screened across 138 drugs.

One of the major hurdles in assessing the reproducibility of this study is its scale. In the registered report the authors propose to reproduce a single finding that was presented in one of the main figures of the original report. While this finding is indeed an important part of the original study it does not embody a large part of the original conclusions: The original paper demonstrated that a drug screen performed across a large collection of cell lines was able to capture accurately a large number of known clinically relevant drug response biomarkers as well as preferential cancer type sensitivities known to occur in the clinic (for example that *BRAF* mutant colorectal cancers are less responsive to BRAF inhibitor than melanomas). It also showed that genomic modeling of the drug response could yield biological insights into drug mechanism of action (Elastic Net analysis outputs). This was distinct from previous smaller scale efforts (and also valuable for other uses) that had been reported prior to the original report (NCI60 results, for example). The reviewers encourage this Reproducibility Report after addressing the following concerns:

Essential revisions:

1) The Introduction of the registered report should better reflect this challenge and the breadth of the results presented in the original report.

For example: "To confirm their high throughput approach Garnett and colleagues explored one novel interaction…". This is similarly an over simplification. Several results "Confirmed" the high throughput approach capability of capturing clinically and biologically relevant drug responses: Chiefly a large number of known gene-drug interactions for clinically active drugs.

End of Introduction: It should be further clarified that the clinical results reported for Olaparib in Ewing's sarcoma correspond to single agent olaparib only. Combinations with olaparib are currently explored but no results have been made public.

Technical aspects:

2) Protocol 1; Procedure; 1: Authors should employ duplicate plates.

3) Protocol 1; Procedure; 4a: The assay (fix and stain) should be stopped when the control (untreated plates) contain at least 100 colonies (ideally over 200).

4) Protocol 1; Known differences from the original study: It is not clear why the authors have chosen not to include all the controls that were presented in the original study. Positive and negative control in a consistent disease background (breast) of *BRCA1/2* deficient and proficient cell lines are important to show that the assay is correctly capturing differential sensitivity to PARP inhibitors across genotypes.

5) Protocol 2, first paragraph: As PARP inhibitor sensitivity relies on replication/proliferation of the cells mechanistically it will be important to show that all cell lines are in good health and proliferative in no drug condition. This is particularly important for MPCs that can be more challenging than average cell lines to maintain in culture. The protocol should include a test for appropriate proliferation.

6) Protocol 2; Procedure; 1: Related to the point above: All assays should include a measurement of proliferation to show that drug treatment occurred while cells were replicating since PARP inhibitor sensitivity depends on replication. Furthermore, differential replication across lines can yield over or underestimate of sensitivity.

7) Protocol 2; Procedure; 6: What level of knock down would be deemed sufficient to declare that the gene expression was affected but no biological effect observed?

8) It is unfortunate that some of the cell lines originally used in Garnett et al. were not available in this Reproducibility Project. However, it will be very useful to the scientific community that this Reproducibility Project and the related reagents and cell lines will be made available for other researchers to reproduce this work.

9) Regarding the statistical analyses, the authors should be aware that repeating the experiment twice or three times on the same cell lines will not give completely independent results which may impact on the results. However, it seems that the power calculations (at least for protocol 1) are conducted using only one run of the experiment. Secondly the power calculation assumes that the observed results will be as strong as those seen in Garnett et al. Even for a real finding, this may be optimistic due to the large number of tests carried out by Garnett and the "winner's curse", i.e. the fact that the most striking findings in a multiple testing context tend to be upwardly biased.

---

## [Author Response]

*Essential revisions:*

*1) The Introduction of the registered report should better reflect this challenge and the breadth of the results presented in the original report.*

For example: "To confirm their high throughput approach Garnett and colleagues explored one novel interaction…". This is similarly an over simplification. Several results "Confirmed" the high throughput approach capability of capturing clinically and biologically relevant drug responses: Chiefly a large number of known gene-drug interactions for clinically active drugs.

Thank you for this suggestion. We have revised the beginning of the Introduction to highlight the large undertaking as well as the utility of the approach and the accurate capture of known associations. The sentence in the first paragraph of the Introduction has also been revised.

End of Introduction: It should be further clarified that the clinical results reported for Olaparib in Ewing's sarcoma correspond to single agent olaparib only. Combinations with olaparib are currently explored but no results have been made public.

We have revised this line to reflect this important aspect.

*Technical aspects:*

2) Protocol 1; Procedure; 1: Authors should employ duplicate plates.

We intended to perform the experiment in duplicate and have revised the manuscript to better reflect this.

3) Protocol 1; Procedure; 4a: The assay (fix and stain) should be stopped when the control (untreated plates) contain at least 100 colonies (ideally over 200).

Thank you for clarifying the sufficient number of colonies in the vehicle treated plates. We have revised the manuscript to reflect this.

4) Protocol 1; Known differences from the original study: It is not clear why the authors have chosen not to include all the controls that were presented in the original study. Positive and negative control in a consistent disease background (breast) of BRCA1/2 deficient and proficient cell lines are important to show that the assay is correctly capturing differential sensitivity to PARP inhibitors across genotypes.

The DoTc2-4510 cell line (uterus tissue that is mutant for *BRCA2* and wild-type for *BRCA1*) was already included as a positive control and we have added the MES-SA cell line (uterus tissue that is wild-type for *BRCA1* and *BRCA2*) as a negative control in the revised Registered Report.

5) Protocol 2, first paragraph: As PARP inhibitor sensitivity relies on replication / proliferation of the cells mechanistically it will be important to show that all cell lines are in good health and proliferative in no drug condition. This is particularly important for MPCs that can be more challenging than average cell lines to maintain in culture. The protocol should include a test for appropriate proliferation.

Thank you for this comment. We have included additional measurements at 24 and 48 hr after treatment (in addition to the originally planned 72 hr measurement) for cells not treated with drug. This will occur for the seeding optimization (Step 1) and for each replicate of the assay (Steps 2-7).

6) Protocol 2; Procedure; 1: Related to the point above: All assays should include a measurement of proliferation to show that drug treatment occurred while cells were replicating since PARP inhibitor sensitivity depends on replication. Furthermore, differential replication across lines can yield over or underestimate of sensitivity.

Thank you for this comment. We have added a test for proliferation for each biological replicate in the revised manuscript.

7) Protocol 2; Procedure; 6: What level of knock down would be deemed sufficient to declare that the gene expression was affected but no biological effect observed?

We do not have information about what level of knock down is necessary in order to observe a biological effect in this assay. The original study does not indicate a threshold, but does report the level of knock down (Supplemental Figure 20). Whether the replication attempt is capable of achieving the same degree of knockdown is important to consider as is the ability to observe a biological effect. Instead of declaring a threshold of knock down we plan to compare our gene expression levels to those of the original study as well as the viability assay results. This will allow the research community to assess this question.

8) It is unfortunate that some of the cell lines originally used in Garnett et al. were not available in this Reproducibility Project. However, it will be very useful to the scientific community that this Reproducibility Project and the related reagents and cell lines will be made available for other researchers to reproduce this work.

We agree and for any reagent or cell line not already available to the research community through a commercial supplier or repository we will work to make these valuable reagents available for other researchers.

*9) Regarding the statistical analyses, the authors should be aware that repeating the experiment twice or three times on the same cell lines will not give completely independent results which may impact on the results. However, it seems that the power calculations (at least for protocol 1) are conducted using only one run of the experiment. Secondly the power calculation assumes that the observed results will be as strong as those seen in Garnett et al. Even for a real finding, this may be optimistic due to the large number of tests carried out by Garnett and the "winner's curse", i.e. the fact that the most striking findings in a multiple testing context tend to be upwardly biased.*

We agree that while repeating the experiment multiple times on the same cell lines does not give complete independence. However, we plan to perform the experiment on several independent samples derived from the same population of cells. While not complete independence, it is as much as can be obtained for a given cell line based experiment – similar to using multiple mice all of which have the same inbred genetic background. However, for Protocol 1 the cell line is the biological replicate, opposed to the random sample from a given cell line. The power calculations were conducted to determine how many Ewing’s sarcoma cell lines and how many osteosarcoma cell lines are necessary to conduct the proposed test. Since each cell line is independent of each other, the number of cell lines of a given disease type were determined to achieve at least 80% power.

Regarding the approach used for the power calculations, we agree there are approaches one could take to guard against inflated effect sizes, such as utilizing the 95% confidence interval of the effect size. However, the Reproducibility Project: Cancer Biology is designed to conduct replications that have 80% power to detect the point estimate of the originally reported effect size. While this has the limitation of being underpowered to detect smaller effects than what is originally reported, this standardizes the approach across all studies to be designed to detect the originally reported effect size with at least 80% power.